# Lightweight Predictive 3D Gaussian Splats

**Junli Cao**[1,2]     **Vidit Goel** [2]     **Chaoyang Wang**[2]     **Anil Kag**[2]     **Ju Hu**[2]
**Sergei Korolev**[2]     **Chenfanfu Jiang**[1]     **Sergey Tulyakov**[2]     **Jian Ren**[2]
[1]University of California, Los Angeles     [2] Snap, Inc.

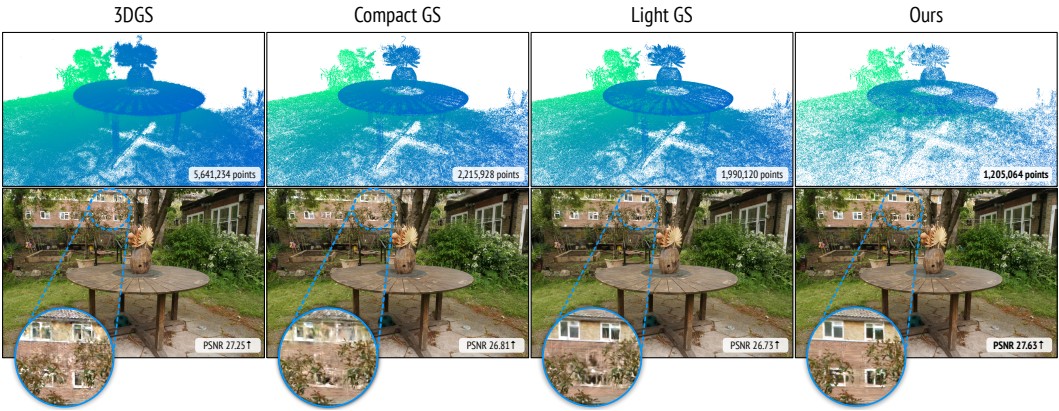

Figure 1: *Top*: We show point clouds of the `Garden` scene (Barron et al., 2022a) obtained using different methods, where we feature the smallest number of points to store. *Bottom*: Images rendered using the compared methods. Ours shows the best PSNR. We magnify a region highlighted with blue, showing that despite significantly smaller storage requirements, we achieve the highest fidelity and can reconstruct the detailed structure of the image. *Zoom-in for greater detail.*

## Abstract

Recent approaches representing 3D objects and scenes using Gaussian splats show increased rendering speed across a variety of platforms and devices. While rendering such representations is indeed extremely efficient, storing and transmitting them is often prohibitively expensive. To represent large-scale scenes, one often needs to store millions of 3D Gaussian, which can occupy up to gigabytes of storage. This creates a significant practical barrier, preventing widespread adoption on resource-constrained devices. In this work, we propose a new *representation* that dramatically reduces the hard drive footprint while featuring similar or improved quality when compared to the standard 3D Gaussian splats. This representation leverages the inherent feature sharing among splats in the close proximity using a hierarchical tree structure, with which only the *parent* splats need to be stored. We present a method for constructing tree structures from naturally *unstructured* point clouds. Additionally, we propose the *adaptive tree manipulation* to prune the redundant trees in the space, while spawn new ones from the significant *children* splats during the optimization process. On the benchmark datasets, we achieve $20\times$ storage reduction in hard-drive footprint with improved fidelity compared to the vanilla 3DGS and $2\times$-$5\times$ reduction compared to the exiting compact solutions. More importantly, we demonstrate the practical application of our method in real-world rendering on *mobile devices* and *AR glasses* in our Webpage.

## 1 Introduction

Gaussian Splatting (3DGS)-based methods are taking the graphics and vision communities by a storm (Luiten et al., 2023; Wu et al., 2023; Yang et al., 2023). They strike the right balance between high-fidelity rendering, fast convergence, and efficient inference (Kerbl et al., 2023). The latter two benefits make 3DGS-based methods superior to Neural Radiance Fields (NeRFs)-based

techniques (Mildenhall et al., 2020; Martin-Brualla et al., 2021; Barron et al., 2022b). Indeed, while NeRFs (Barron et al., 2022a) show high-fidelity renderings too, apart from several exceptions (Cao et al., 2023; Wang et al., 2022; Chen et al., 2023b; Müller et al., 2022a), their training and inference time is often prohibiting real-time and edge-based applications. 3DGS-based approaches represent a 3D scene using an explicit, point-based representation (Aliev et al., 2020). The 3D Gaussians are efficiently rasterized to 2D images, with much faster rendering than neural volumetric rendering approaches (Kerbl et al., 2023). However, to represent sophisticated geometry and texture, especially for large-scale scenes, a significant amount of splats along with their attributes need to be stored, which can amount to even gigabytes of storage.

In a world of connected devices, real-time experiences and applications, this storage requirement imposes a heavy toll on the hard-drive and the transmission bandwidth. Hence, several initial solutions have been proposed to reduce the storage for 3DGS, such as incorporating a sparse voxel grid (Lu et al., 2023) or applying more aggressive pruning of the 3D points (Fan et al., 2024; Lee et al., 2024). Yet, existing studies still suffer either from large storage requirements (Lu et al., 2023) or inferior rendering quality compared to 3DGS (Fan et al., 2024; Lee et al., 2024).

In this work, we present a lightweight hierarchical Gaussian splats representation that takes advantage of the spatial relationships among unstructured and isolated splats, offering improved rendering quality while significantly reducing storage requirements. Intuitively, splats in close proximity exhibit similar geometry and texture. Therefore, we leverage feature sharing among nearby splats and propose structuring them into a hierarchical tree, where the *parent* splats are employed to neural-predict splats that share similar features. We call these neural-predicted splats the *children* splats. Note that *children* splats do not have to be stored and can be neural-predicted on-the-fly instead. We use hash-grid (Müller et al., 2022b) to encode the offsets that are used to estimate the 3D locations of *children* splats. In addition, within the same hash grid, we first query the features of both the *parent* and *children* splats and apply an attention-based mechanism to attend to them. This attention is crucial for facilitating feature sharing within the tree. The attended features are then input into a shallow MLP to predict the Gaussian attributes. We opt for the hash-grid due to its ability to facilitate feature sharing in close proximity by interpolating spatially adjacent feature vectors. Our representation is *independent* of grid-based structures; any representation that encourages feature sharing can be utilized (*e.g.*, K-plane (Fridovich-Keil et al., 2023)).

To build such tree structures, we first allow every point obtained from SfM to be considered as a *parent* splat, and be used to predict its *children* splats. Since the splats in our representation are structured and treated as a cohesive unit, we further introduce the *Adaptive Tree Manipulation*(ATM) module to manage the tree during the optimization process. Specifically, we do not impose a limit on the depth of the tree. This means that a *children* splat can serve as a *parent* in the next optimization iteration and has its own *children* splats if it is deemed significant. Additionally, insignificant *parent* splats are pruned along with their insignificant *children*. Note that significant *children* are promoted to *parent* regardless of the significance of their *parent*. For instance, an insignificant *parent* may be removed in the next optimization iteration, but it can still have significant *children* that are promoted to *parent*. This flexible tree manipulation enables certain areas with complex geometry to include more splats for more accurate modeling.

Fig. 1 shows the Garden scene (Barron et al., 2022a) reconstructed by the standard Gaussian Splats (Kerbl et al., 2023), Compact GS (Lee et al., 2024), Light GS (Fan et al., 2024) and the proposed approach. First, we observe a significantly reduced density of points in the point cloud reconstructed by our approach. This, and the predicting of the attributes instead of storing them, significantly reduces the storage requirement for our method. Second, we show improved PSNR scores and visual quality, when we zoom-in into the details of the rendered images. We summarize our contributions as follows:

1. We propose a hierarchical tree structure to model the inherent spatial relationships among splats and an attention mechanism to enhance the relationship within the hierarchy.
2. We propose Adaptive Tree Manipulation in conjunction with the hierarchical representation to effectively refine the tree for improved modeling.
3. Our representation achieves $20\times$ reduction on average in hard-drive footprint, with improved PSNR and comparable SSIM and LPIPS comparing to 3DGS and $2\times$-$5\times$ storage reduction comparing the exiting works. Additionally, we showcase the practical real-world rendering applications of our method on mobile devices and AR glasses.

## 2 RELATED WORK

**Novel View Synthesis.** Research on rendering scenes from unseen viewpoints with photorealism has evolved over several decades (Greene, 1986; Chen & Williams, 2023; Levoy & Hanrahan, 2023; Buehler et al., 2023; Srinivasan et al., 2019). Traditional approaches typically rely on explicit depth estimation to warp pixels for generating novel views (Kalantari et al., 2016; Penner & Zhang, 2017; Choi et al., 2019; Riegler & Koltun, 2021). However, the accuracy of depth estimation algorithms is critical, and handling disocclusions during rendering adds complexity. An alternative approach involves Multiplane Images (MPI) (Zhou et al., 2018; Srinivasan et al., 2019; Flynn et al., 2019), which learn a representation associating objects within the scene with fronto-parallel layers. This structured representation facilitates efficient rendering from different viewpoints while preserving depth relationships and occlusions. More recently, Neural Radiance Fields (NeRF) (Mildenhall et al., 2020) have gained popularity for their ability to achieve highly realistic rendering, even in scenarios involving complex view-dependent lighting effects such as transparency and reflectance. However, the weakness of NeRF lies in its volumetric rendering formulation, which necessitates sampling a large number of points per ray to render a single pixel. This high computational cost limits the usage of NeRF for real-time or on-device applications. While efforts to reduce computational requirements for volumetric rendering have been a focus of recent research (Liu et al., 2020; Neff et al., 2021; Garbin et al., 2021; Reiser et al., 2021; Lindell et al., 2021; Yu et al., 2021; Müller et al., 2022a; Fridovich-Keil et al., 2022; Lombardi et al., 2021; Cao et al., 2023; Gupta et al., 2024), point-based rendering, particularly 3D Gaussian Splatting (3DGS) (Kerbl et al., 2023), presents another promising direction for real-time view synthesis.

**Efficient Representation for 3D Gaussian Splatting.** Despite the benefits of 3DGS (Kerbl et al., 2023), the disadvantages of bulky storage are noteworthy. As a result, several approaches (Fan et al., 2024; Lee et al., 2024; Lu et al., 2023; Girish et al., 2023; Niedermayr et al., 2024; Morgenstern et al., 2023; Navaneet et al.) have been proposed for compressing 3DGS. Several compression techniques have been explored such as pruning the redundant Gaussian (Fan et al., 2024; Lee et al., 2024) and utilizing codebooks (Fan et al., 2024; Lee et al., 2024; Niedermayr et al., 2024; Navaneet et al.). LightGS (Fan et al., 2024) introduces a point pruning and recovery process to minimize redundancy in Gaussian splats, utilizes distillation and pseudo-view augmentation to distill spherical harmonics to a lower degree, and employs quantization to further reduce storage. While LightGS achieves considerable storage reduction, it results in noticeable fidelity degradation compared to the original Gaussian splatting due to quantization. CompactGS (Lee et al., 2024) proposes using a grid-based neural field to implicitly represent view-dependent colors rather than explicitly storing spherical harmonics per point, offering promising storage efficiency without significant fidelity loss. Eagles (Girish et al., 2023) utilizes quantized embedding to quantize the per-point attributes and pruning strategy to remove redundant Gaussian, leading to lower storage memory. ScaffoldGS (Lu et al., 2023) exploits the spatial feature sharing by distributing local splats using anchor points, reparameterizing splats positions relative to these anchors to enable anchor-based point growing and pruning strategies for redundancy reduction in 3DGS. Our method shares similarities with CompactGS (Lee et al., 2024) and ScaffoldGS (Lu et al., 2023) while exhibits crucial *differences*. *First*, unlike CompactGS, we utilize a combination of neural fields and self-attention layers to predict not only view-dependent colors but also geometric properties. *Second*, in contrast to both approaches which explicitly store the position of every splat in the point cloud, our method only stores a small subset of splats, referred to as *parent*, while predicting the remaining points on-the-fly during rendering. This substantially reduces memory footprint. *Third*, anchor-based representation essentially creates a tree structure but restricts the depth of the tree to one. The growth strategy focuses solely on the anchor points, neglecting the growth directly from the splats. In contrast, our hierarchical tree representation combined with the proposed ATM take into account the significance of both *parent* and *children* splats, allowing for a strategy of sub-tree expansion.

## 3 PRELIMINARIES

3D Gaussian Splatting (3DGS) (Kerbl et al., 2023) represents a scene with 3D points **x**. The points are initialized with a coarse point cloud obtained using Structure-from-Motion (SfM) (Schonberger & Frahm, 2016). These Gaussians, $G(\mathbf{x})$, serve as the anisotropic volumetric splats defined by their position (mean $\mu$) and 3D covariance ($\Sigma$) as $G(\mathbf{x}) = e^{-\frac{1}{2}(\mathbf{x}-\mu)^T \Sigma^{-1}(\mathbf{x}-\mu)}$. To ensure $\Sigma$ remaining

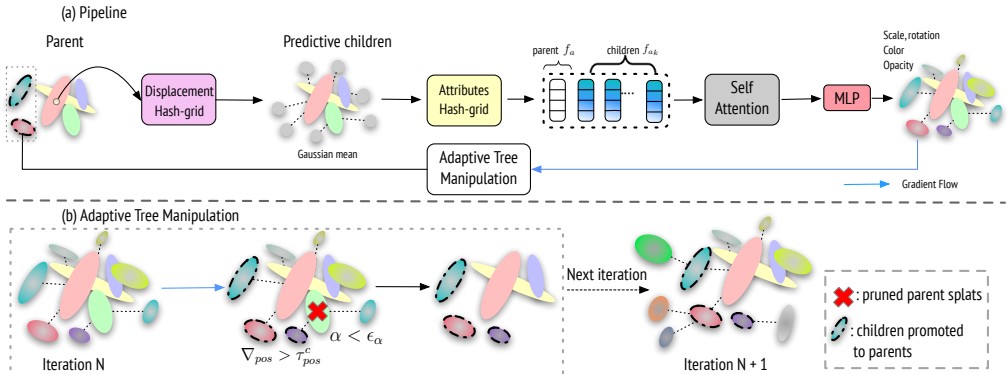

Figure 2: *Top*: Overview of our tree construction pipeline with the initial *parent* splats derived from SfM. The *children* spats are inferred **on-the-fly** from the *parent* splats via querying the displacement hash-grid. To estimate the Gaussian attributes like scale, rotation, color, and opacity, attribute features $f_a$ and $f_{ak}$ obtained **on-the-fly** from the attributes hash-grid are aggregated with self-attention. *Bottom*: Tree manipulation through ATM. The significant *children* splats are promoted to *parent* (regardless of the status of their *parent*, *e.g.*, pruned) such that they have their own *children* in the next iteration. Bad trees (*e.g.*, transparent *parent*) are removed together with the insignificant *children*.

positive semi-definite during optimization, it is represented with an equivalent yet effective formulation with the scaling matrix $S$ and the rotation matrix $R$, such that $\Sigma = RSS^T R^T$. The attributes of the 3D splats (*e.g.*, location, covariance, and opacity) together with the directional appearance of the radiance filed, represented via the spherical harmonics (SH) (Sara Fridovich-Keil and Alex Yu et al., 2022), are end-to-end learned using optimization.

To render an image, 3D $G(\mathbf{x})$ are first transformed into 2D Gaussians (denoted as $G'(\mathbf{x})$) (Zwicker et al., 2001). 3DGS uses an efficient tiled-based rasterizer that presorts primitives for the entire image, allowing fast $\alpha$-blending of anisotropic splats. The color $C$ of a pixel is computed by blending $N$ 2D Gaussians that overlap at the pixel as: $C = \sum_{i \in N} c_i \alpha_i G_i'(\mathbf{x}) \prod_{j=1}^{i-1}(1 - \alpha_j G_j'(\mathbf{x}))$, where $c_i$ represents view-dependent colors for each splat, $\alpha_i$ is the opacity. With the highly optimized rasterizer for modern GPUs, 3DGS render high-fidelity scenes in real-time across many platforms. These benefits come with a cost. 3DGS require a significant number of 3D Gaussians, sometimes needing gigabytes for complex large-scale scenes. This requirement limits their application on edge devices, as downloading gigabytes over the network and storing them is hardly feasible or practical.

# 4 METHOD

We show a high-level overview of our approach in Fig. 2. Our primary motivation is to use a hierarchical representation(*i.e.*, tree) to model the spatial relationships among the splats. We show that the locations of *children* splats and associated attributes —position, color, scale, *etc.*— can be derived from the *parent* using a small neural network. This allows us to store only the *parent* splats along with the weights of the neural network. To achieve this, we initially represent a 3D scene as a forest of depth-1 tree structures where the *parent* splats are initialized from SfM (Schönberger & Frahm, 2016) and the *children* are neural-predicted on-the-fly. The trees are then refined and expanded to sub-trees during the optimization process using *Adaptive Tree Manipulation*. Formally, we represent a scene using $\mathcal{S} = \{\mathcal{X}_1, \mathcal{X}_2, \dots \mathcal{X}_n\}$, where $\mathcal{X}_i$ is tree and each node contains the attributes, such as position $(x)$, color $(c)$, opacity $(\alpha)$, scale $(s)$, and rotation $(r)$. This representation can be stored very efficiently, as for each tree we need to save only the positions and scales of *parent* splats and small neural network shared across the trees, to predict all the other attributes of the tree.

## 4.1 NEURAL REPRESENTATION FOR LIGHTWEIGHT PREDICTIVE SPLATS

We model close relationship between a parent and children nodes. Specifically, we assume that the children nodes are in the vicinity of parent node and have similar geometric and appearance attributes such as shape, color and opacity. We satisfy these requirements by using a hash-grid based approach (Müller et al., 2022b; Chen et al., 2023a) as our representation, which has an inherent

property to return similar features when queried with the points located nearby via feature interpolation. Below we describe how a tree ($\mathcal{X}_i$) can be represented in storage efficient manner. In what follows we drop the index $i$. We use the notation *node* to refer the splat in the context of a *tree*.

For a hash-grid $\mathcal{H}(\cdot)$ shared across the trees and *parent* node positions $x_p$, we query the features as $f = \mathcal{H}(x_p)$ and use them to predict the displacement of *children* and attributes of the tree. We divide $f$ into two halves $f \equiv \{f_\Delta \in \mathbb{R}^{D/2}, f_a \in \mathbb{R}^{D/2}\}$, where the first half ($f_\Delta$) represents displacement and is used to predict the position of children. The second half ($f_a$) is used to predict other attributes.

**Predicting Position.** We want *children* and *parent* nodes to represent similar geometry and appearance. Hence, *children* should be located in the vicinity of the *parents* nodes. We model the position of *children* as their displacement from their *parent* nodes. For the *parent* we predict the position of $k^{\text{th}}$ child using $x_k = x_p + g_{\text{pos}}(f_\Delta)[k]$ where $g_{\text{pos}}$ is an MLP with output shape $K \times 3$.

Having the positions of all nodes in the tree, we can predict the rest of the attributes, such as scale, rotation, color, and opacity. We reuse the hash-grid to get the attribute feature ($f_{ak}$) for $k^{\text{th}}$ child node using $\mathcal{H}(x_k)$. A naive approach to extract the remaining attributes using $f_a$ and $f_{ak}$ is to pass the latter to an MLP get scale, rotation, color and opacity. We found such approach to be sub optimal. A hash-grid representation implicitly makes the representation of spatially points similar. There is no mechanism to share information between the features after they are computed. Since there is relation between physical attributes of the *parent* and *children* nodes, having such information sharing mechanism is beneficial. To this end, we propose a modified self-attention mechanism to better capture the inter-dependencies between *children* and *parent* nodes. To do so, we first obtain the aggregated feature $\mathcal{F}_a \in \mathbb{R}^{K+1 \times D/2}$ by concatenating features of all the nodes in the tree, such as $\mathcal{F}_a = \texttt{Concat}(\{f_a, (f_{a1}, \ldots, f_{aK}\})$, where $\texttt{Concat}$ is a concatenation operation. We then apply a modified self-attention operation on $\mathcal{F}_a$ to get the final feature $\mathcal{F}'_a$:

$$\mathcal{F}'_a = \mathcal{F}_a + \lambda \sigma \Big( \frac{\mathcal{P}_1(\mathcal{F}_a) * \mathcal{P}_2(\mathcal{F}_a)^T}{\sqrt{d}} \Big) * \mathcal{F}_a, \tag{1}$$

where $\sigma(\cdot)$ is a Softmax function, $\mathcal{P}_i(\cdot)$ is a projection matrix, $d$ is a scaling factor set as $D/2$, $\lambda$ is a hyper-parameter for balancing the information trade-off from the attention mechanism and $*$ denotes the matrix multiplication. Different from vanilla attention (Vaswani et al., 2017), we do not apply positional embedding, so that Eq. 1 is permutation invariant which is an important property to maintain while working with point clouds (Qi et al., 2016). Further, we use the unprojected $\mathcal{F}_a$ when multiplying with $\sigma(\cdot)$, since we empirically found no performance gain by projecting $\mathcal{F}_a$. Next, we split $\mathcal{F}'_a$ in $K+1$ attribute feature vector to predict the remaining attributes for each node in the tree.

**Predicting Scale and Rotation.** It is vital to properly initialize the scale of Gaussians for stable training. For instance, Gaussians with small scales make minimal contributions to the rendering quality, mainly because of their limited volume. In contrast, large Gaussians can potentially contribute to every pixel during rasterization, leading to a significant amount of GPU memory. Hence, to make training stable and minimize storage needs at the same time, we adopt a middle-ground strategy. More specifically, we represent the scales of *children* as a scaled version of their *parents* ($s_p$): $s_k = \hat{s}_k s_p$ where $\hat{s}_k$ is predicted by an MLP. In case of rotation, we directly regress it for both *parents* and *children* nodes using the corresponding attribute feature vector. We share the weights of the MLP to regress both scale and rotation. We experimentally found, that including position of node ($x_k$), the distance of the point to the center of the axis aligned bounding box ($b_k$) along with attribute feature (($f'_{ak}$) improves performance: $\hat{s}_k, r_k = g_{rs}(f'_{ak}, x_k, b_k)$.

**Predicting Color and Opacity.** 3DGS uses degree-3 spherical harmonics (SH) for view-dependent color representation (Kerbl et al., 2023). However, we find such design is unnecessary and the color can be directly predicted using from feature vectors and a viewing direction. We use an MLP that takes them as an input and directly predicts the color as output, $c_k = g_c(f'_{ak}, d_k)$ where $d_k$ is the viewing direction of the node in the tree. This reduces the storage by a significant amount as previously each splat storing the spherical harmonics individually. To predict opacity, we use another MLP with inputs as $f'_{ak}$ and the position of the node to get corresponding opacity, $o_k = g_o(f'_{ak}, x_k)$.

We described all the operations above for a single tree. The same operation is extended for all the trees. Further, the neural networks for all the operations share their weights across all the trees. To summarize, the proposed representation efficiently represents the tree structure with hash-grid based neural representations $\mathcal{H}(\cdot)$ and a few MLPs. We only store position and scale of parent nodes and the weights of our neural networks, while the rest of the properties is regressed as described above.

## 4.2 ADAPTIVE TREE MANIPULATION

3DGS (Kerbl et al., 2023) starts by using an initial point cloud from SfM. To allow for some flexibility in the point cloud structure they propose several techniques to add and delete the Gaussians during optimization. These techniques work effectively for individual splats, but naively applying them to our tree (*i.e.*, applying directly to the *parent* nodes of a tree) results in sub-optimal or incorrect outcomes(see Tab. 2a and Fig. 8). This is because such an approach implicitly limits the depth of all trees to one, overlooking the significance of the *children* nodes and preventing sub-tree expansion and consequently hindering the quality. This consideration necessitate a new strategy that incorporates the importance of the *children* into the process. On the other hand, a structural representation facilitates feature sharing, while simultaneously entangling the *parent* and *children* nodes into a cohesive unit due to the nature of the tree. Consequently, operations on the *parent* directly affect the *children* (*e.g.*, pruning the *parent* results in the removal of the *children*). Considering these factors, we propose the following strategy (see Fig. 2):

- **Promotion** A *children* can become a *parent* if it is deemed significant during training, and it can have its own *children* in the next iteration.
- **Pruning** Removing or pruning the *parent* operates only on itself and its insignificant *children*. Significant *children* are unaffected.
- **Cloning** Cloning the *parent* operates on the entire tree (*i.e.*, all *children* are cloned as well).

To determine the significance of a *children*, we track the position gradient of all the *children* splats. When the gradient of a child node is above a certain threshold $\tau_{\text{pos}}^c$ (*i.e.*, $\nabla_{pos}^c > \tau_{pos}^c$ ) where $\tau_{pos}^c$ is a hyper-parameter, we consider the *children* significant. Then we promote the *children* node to become a new *parent* in next training iteration (satisfying **Promotion** ). This is crucial to represent complicated regions in the scene where there might not be many *parent* nodes. Once the *children* has been promoted to *parent* node we apply clone and split operations to all the *parents* following similar practices in 3DGS (satisfying **Cloning**). To address **Pruning**, we first check if any nodes meet the criteria outlined in **Promotion** and promote them if needed. Then to delete the tree we only rely on the statistics of *parent* node. This is because we assume if a *children* node was important then it would have been already promoted to become a new *parent*. Hence, we can safely delete the current *parent* that will in turn delete all the corresponding insignificant *children* nodes. Specifically, for deleting the trees we track the scale and opacity of the *parent* nodes and delete them if they are below a certain threshold similar to 3DGS.

## 4.3 TRAINING

Our model, including the hash-grid and MLPs, is *end-to-end* learnable guided by the $\mathcal{L}_1$ loss between the rendered images and ground-truth images along with a D-SSIM loss, such that:

$$\mathcal{L} = (1 - \beta)\mathcal{L}_1 + \beta\mathcal{L}_{\text{D−SSIM}}, \tag{2}$$

where $\beta$ is set as $0.2$ following the setting in  (Kerbl et al., 2023).

We use a warm-up training scheme that helps in convergence of the model (Kerbl et al., 2023). The warm-up consists of training the model in a low resolution setting, eventually moving to higher resolution after a certain number of steps have been completed. We found that the warm-up strategy is crucial to correctly position the splats and densify the regions. Without the warm-up, the model struggles to populate enough splats in the background area, despite the importance of the area resulting in substandard performance. Please refer to Appendix material for details.

## 5 EXPERIMENTS

**Dataset and Metrics.** We evaluate our method using *seven* scenes from the Mip-NeRF 360° dataset (Barron et al., 2022b), *two* scenes from Tank&Temples (Knapitsch et al., 2017), and *two* scenes from Deep Blending (Hedman et al., 2018). We use the widely adopted metrics like PSNR, SSIM (Wang et al., 2004), and LPIPS (Zhang et al., 2018) to assess the quality for image reconstruction. We also report the storage size (in MB) for various methods along with their on-device capabilities. We benchmark the Gaussian Splatting based methods on iPhone 14 with our implementation of the mobile application. We report three configurations of our method named C1, C2, and C3 by varying feature dimension $D$ of the hash-grid $\mathcal{H}$. C1 is our smallest model with $D = 32$

followed by C2 with $D = 48$ and C3 is largest with $D = 64$. Since our framework adds and removes points during optimization, the final storage for each model can vary. For each dataset we report the average size of all the scenes within one configuration. The metrics, too, are averaged over all scenes of each dataset. Per-scene quantitative results are in the Appendix.

**Quality *vs.* Storage**. First, we show that our approach provides a practical means of satisfying diverse technical requirements. We can reduce or increase the feature dimension of the hash-grid and the number of points, while still maintaining similar or superior rendering quality. In Fig. 3, we plot PSNR, evaluated on the dataset introduced by (Barron et al., 2022a), for contemporary models as well as for three configurations of our approach. Our smallest configuration is almost 50% smaller than the smallest prior work (LightGS (Fan et al., 2024)), and shows the same rendering quality. Our largest configuration, which is still 32% smaller than the smallest existing work, shows significantly increased PSNR. To give the reader a better perspective, we also plot conventional works with large hard-drive footprint (Lu et al., 2023; Kerbl et al., 2023). Our largest configuration, which uses only 20% of ScaffoldGS (Lu et al., 2023) and only 4.5% of 3DGS (Kerbl et al., 2023) storage, shows higher quality than both of these much larger works. These advantages of our method are crucial for mobile deployment. Less disk storage also helps in speeding up transmission that significantly impacts user experience when sharing content.

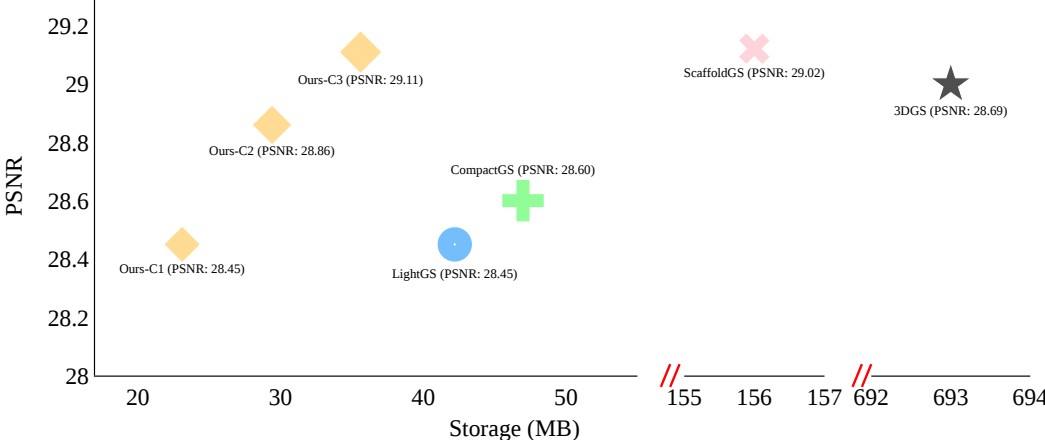

Figure 3: Comparisons of PSNR and storage computed on the dataset from Barron et al. (2022a).

### 5.1 COMPARISON RESULTS

**Quantitative Results**. Tab. 1 shows the quantitative results on real-world scenes, spanning from large-scale urban landscapes to intricate indoor and outdoor environments. We compare our approach against 3DGS (Kerbl et al., 2023) and concurrent works (*i.e.*, LightGS (Fan et al., 2024), CompactGS (Lee et al., 2024), Eagles (Girish et al., 2023), CompGS (Navaneet et al.) and ScaffoldGS (Lu et al., 2023)). On the Mip-NeRF 360° dataset, we achieve the *best* PSNR among all the approaches. Compared with 3DGS (Kerbl et al., 2023), we obtain a significant storage *reduction*, *i.e.*, $19.5\times$, and require $3.5\times$ *fewer* 3D points. On the Tank&Temples (Knapitsch et al., 2017) dataset, although ScaffoldGS (Lu et al., 2023) has better PSNR than our approach, our storage is almost $2.4\times$ *smaller* than ScaffoldGS. Compared with 3DGS (Kerbl et al., 2023) on this dataset, we require $1.9\times$ *fewer* 3D points and $11.3\times$ *less* storage. On the Deep Blending (Hedman et al., 2018) dataset, our method has *higher* PSNR and $19\times$ storage *reduction* than 3DGS (Kerbl et al., 2023).

Fig. 4 demonstrates the high-quality rendering of our method produced using C3 configuration across 5 example scenes covering all the datasets. We see various examples where our method outperforms previous compression works. We can see our models can better capture background details (row 3, 5), better capture reflections (row 2) while being the smallest or of comparable size. It can also capture intricate details where other methods fail such as ceilings (row 1, 4).

**On-Device Capability**. We explore the feasibility of running splatting based methods on mobile devices. We use iPhone14 and Snap AR glasses Spectacles to implement the applications. For fair comparison, we unpack splats from all methods to a standard 3DGS format (Kerbl et al., 2023) for

Table 1: Quantitative comparisons on three widely used benchmark datasets, including Mip-NeRF 360° dataset (Barron et al., 2022a), Tanks&Temples (Knapitsch et al., 2017), and Deep Blending (Hedman et al., 2018). We report the image quality metrics, such as PSNR, SSIM, and LPIPS, and the required storage. We also report the on-device capability of each Gaussian Splatting based work (On-Device in the table), where OOM denotes Out-of-Memory error and ✓ denotes the real-time capability ($> 30$ fps) on our tested device, *i.e.*, iPhone14. and $-$ denotes unknown of on-device capability. The evaluation results on other works are obtained from their papers. Compared with the methods that are capable to run on mobile devices, our models (Ours-C1, C2, C3) can obtain smaller model size with higher rendering quality (*i.e.*, PSNR).

| Method | On-Device | Mip-NeRF 360° Dataset | | | | Tank&Temples | | | | Deep Blending | | | |
|---|---|---|---|---|---|---|---|---|---|---|---|---|---|
| | | PSNR ↑ | SSIM ↑ | LPIPS ↓ | Storage ↓ | PSNR ↑ | SSIM ↑ | LPIPS ↓ | Storage ↓ | PSNR ↑ | SSIM ↑ | LPIPS ↓ | Storage ↓ |
| ScaffoldGS (Lu et al., 2023) | OOM | 29.02 | 0.848 | 0.220 | 156MB | 23.96 | 0.853 | 0.177 | 87MB | 30.21 | 0.906 | 0.254 | 66MB |
| 3DGS (Kerbl et al., 2023) | OOM | 28.69 | 0.870 | 0.182 | 693MB | 23.14 | 0.841 | 0.183 | 411MB | 29.41 | 0.903 | 0.243 | 676MB |
| CompGS (Navaneet et al.) | - | 27.16 | 0.808 | 0.228 | 50.30MB | 23.47 | 0.840 | 0.188 | 27.97MB | 29.75 | 0.903 | 0.247 | 42.77MB |
| Eagles (Girish et al., 2023) | ✓ | 27.15 | 0.808 | 0.238 | 68.89MB | 23.41 | 0.840 | 0.200 | 34MB | 29.91 | 0.910 | 0.250 | 62MB |
| LightGS (Fan et al., 2024) | ✓ | 28.45 | 0.857 | 0.210 | 42.48MB | 22.83 | 0.807 | 0.242 | 22.43MB | - | - | - | - |
| CompactGS (Lee et al., 2024) | ✓ | 28.60 | 0.855 | 0.211 | 46.98MB | 23.32 | 0.831 | 0.201 | 39.43MB | 29.79 | 0.901 | 0.258 | 43.21MB |
| **Ours-C1** | ✓ | 28.45 | 0.837 | 0.235 | 23.40 MB | 23.19 | 0.810 | 0.239 | 22.00 MB | 29.32 | 0.895 | 0.282 | 22.90MB |
| **Ours-C2** | ✓ | 28.86 | 0.851 | 0.217 | 29.50 MB | 23.47 | 0.820 | 0.228 | 29.05MB | 29.61 | 0.896 | 0.277 | 29.15MB |
| **Ours-C3** | ✓ | 29.11 | 0.857 | 0.210 | 35.60MB | 23.82 | 0.829 | 0.210 | 35.32MB | 29.89 | 0.902 | 0.267 | 35.40MB |

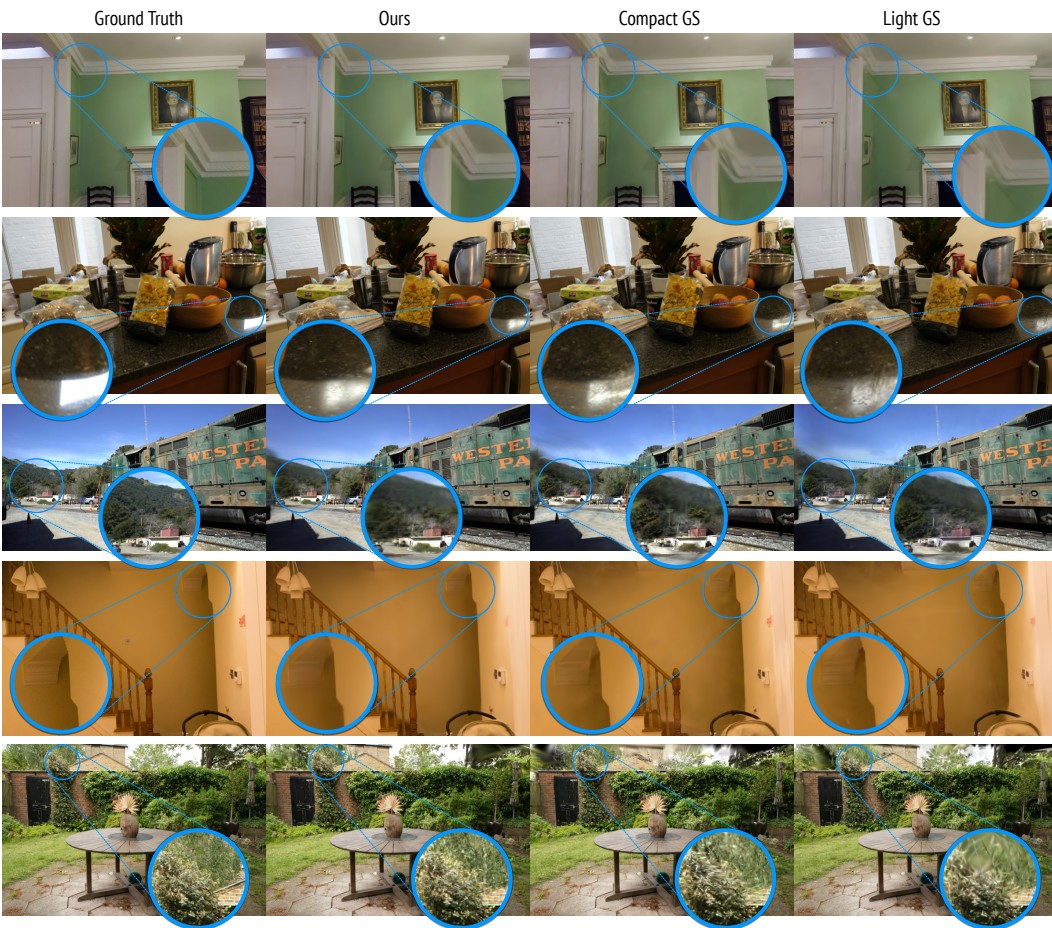

Figure 4: Visual comparisons with methods offering efficient GS representations (Lee et al. (2024); Fan et al. (2024)). We magnified regions to show qualitative differences. Our approach (C3) can render images with high-quality while greatly saving the storage. *Zoom-in for greater detail*.

rendering. We observe Out-of-Memory error when running all scenes from the three benchmark datasets for 3DGS and ScaffoldGS, owing to their large number of splats. Our method can successfully run on device (as in Fig. 5), and achieves smaller and better rendering quality compared to LightGS (Fan et al., 2024), CompactGS (Lee et al., 2024) and Eagles (Girish et al., 2023). Check our Webpage for video demos.

Table 2: Ablation analysis on core design elements. We report the PSNR for each experiment.

(a) **Ablation Study on the propose contributions.** FE stands for frequency encoding. **Full** refers to the full settings.

|  | FE | w/o Agg. | w/o ATM | w/o Contract. | **Full** |
|---|---|---|---|---|---|
| Bicycle | 19.72 | 22.80 | 22.66 | 22.88 | 23.68 |
| Playroom | 23.36 | 28.65 | 28.74 | - | 29.27 |

(b) **Analysis of self-attention.** We use the attention with different number of heads (H) and attention head dimension (F), and different $\lambda$ in Eqn. 1.

|  | H1-F32 | H2-F16 | H4-F8 | $\lambda$=0.1 | $\lambda$=0.5 | $\lambda$=1.0 |
|---|---|---|---|---|---|---|
| Bicycle | 23.68 | 23.31 | 23.20 | 23.55 | 23.68 | 23.53 |
| Playroom | 29.27 | 29.15 | 29.10 | 29.14 | 29.27 | 29.22 |

**Storage Analysis**. The storage of our method consists of four components: the hash-grid, *parent* splat locations and scales, and MLPs. The hash-grid is stored in 8 bits, while the other components are stored in 16 bits. Notably, the hash-grid occupies half of the total storage. However, our representation is independent of feature encoding module (*i.e.*, hash-grid encoding) thus it allows for easy adjustments to more efficient representations if necessary. For instance, in our C1 configuration using the dataset Barron et al. (2022a), the storage allocation for each component is as follows: 12.5 MB for the hash-grid, 5.2 MB for the *parent* splat locations, 5.2 MB for the scales, and 0.5 MB for the MLPs.

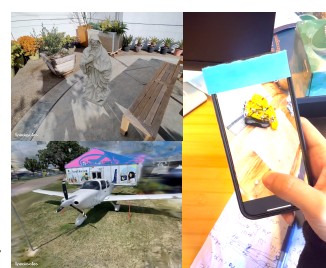

Figure 5: Demo of running our method on iPhone 14 and Snap AR glasses Spectacles.

**Inference Phase Optimization**. Predicting the attributes brings overheads during the inference. However, we note that only the attribute color is view-dependent and the rest remain the same for all frames. Therefore, to minimize the computation cost, we opt to run the color prediction (small MLPs) only and the rest of the attributes are retrieved from the first frame. Thus, we get real-time rendering on the mobile phones and achieve comparable speed as 3DGS (Kerbl et al., 2023). We report two large-scale complex scenes Bicycle and Garden from Barron et al. (2022a) on Nvidia A100: the rendering FPS are 61, 55 for our method and 77, 63 for 3DGS, respectively.

## 5.2 ABLATION ANALYSIS

We perform comprehensive analysis on various components of our methods using our C3 configuration. Here we choose two representative scenes to perform experiments: one unbounded outdoor scene Bicycle from Mip-NeRF 360° dataset (Barron et al., 2022a) and one indoor scene Playroom from the Deep Blending dataset (Hedman et al., 2018). We report the best PSNR that is achieved within 10K steps for all experiments.

**Importance of Hash Grid.** We replace the hash grid with the frequency encoding of the 3D position followed by a 2-layer MLP to output a $D = 64$-dimensional feature vector, which has the same dimension as the one from hash grid. We denote the setting as *FE*. Without hash-grid we see a significant drop in the performance, highlighting the importance of the feature alignment encoded within the spatial hash grid.

**Importance of Attention Mechanism.** When we remove self attention mechanism between the nodes of the tree, it is detrimental to the performance (Tab. 2a w/o Attn). This validates our motivation that there is relation between various physical attributes of the nodes of tree hence there needs to be a mechanism to facilitate the sharing of information. Additionally, adding attention mechanism reduces the number of *parent* splats making our method storage efficient: 884K v.s 1.06M averaged across all scenes in dataset (Barron et al., 2022a). We hypothesise that a configuration with attention can pull information from nearby splats, allowing the method to reduce the number of splats to store and represent the scene efficiently. Additionally, we ablated various configurations by varying $\lambda$ in Eq. 1 and the number of heads in attention to find the best configuration (Tab. 2b). We see a right balance between the input features and attention features is important for best performance.

**Adaptive Tree Manipulation (ATM).** We remove Adaptive Tree Manipulation (ATM) and add or delete the trees based only on *parent* nodes statistics and observe a drop in PSNR (Tab. 2a) also visible in rendered images Fig. 8. This is because there is no mechanism to promote important *children* to *parent* that might hinder in populating trees correctly and failing to represent complex

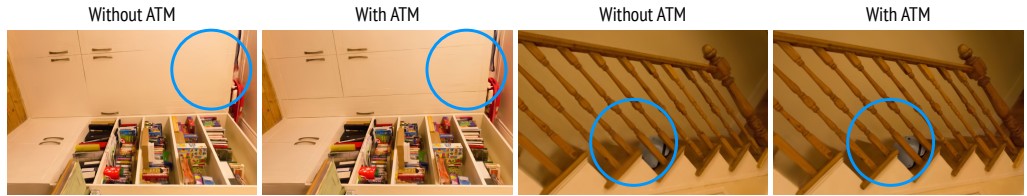

Figure 6: Visual comparison of model trained with and without ATM. We can see that model trained without ATM fails to model intricate details in the scene.

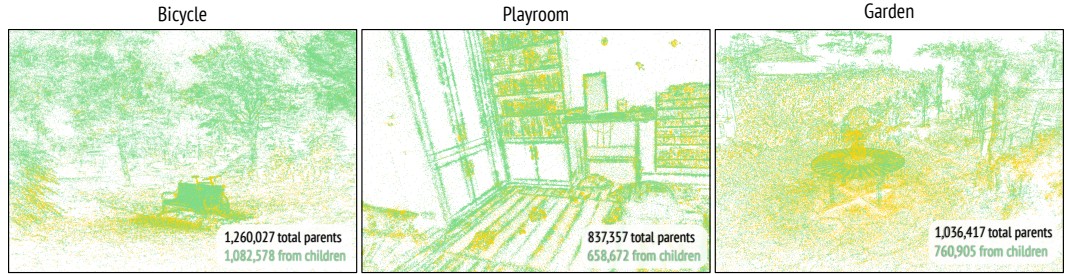

Figure 7: The effect of Adaptive Tree Manipulation (ATM). Yellow points indicate the splats who have not changed the *parent* status during entire optimization. Green points represent former *children* that have been promoted to *parents*. Around 80% of *parents* are from our ATM operation.

scenes effectively. On the other hand, this might also lead to deletion of important *children* nodes when deleting a parent. Our proposed ATM method can effectively alleviate these issues.

Additionally, in Fig. 7 we show the point clouds of three scenes. Green points represent *parents* promoted from *children* during the optimization. Yellow points show *parents* that stayed *parents* during entire optimization. It is clearly seen that the majority of the *parent* nodes are formed by promoting *children* nodes. Further note that parts with relatively flat geometry exhibit more yellow, while sophisticated geometry with high frequency details contain more green. Hence, ATM brings a further benefit of being able to fit sophisticated geometry better.

**Inputs of MLP for attribute prediction.** Tab. 3 shows the analysis for the inputs used to predict the attributes. We conduct the experiments of without using the distance from points to the center of AABB (denoted as *w/o Distance*) and without using the 3D position information (denoted as *w/o Position*) to predict attributes. Position is very crucial for

Table 3: Analysis of the inputs used for attributes prediction.

|  | w/o Distance | w/o Position | SH D1 | SH D2 | SH D3 |
|---|---|---|---|---|---|
| Bicycle | 23.32 | 9.72 | 23.05 | 23.60 | 23.68 |
| Playroom | 29.18 | 6.19 | 29.23 | 29.15 | 29.27 |

training while distance further improves the performance. We also analyze the degrees of the SH encoding on the view directions by performing degree from 1 to 3 (denoted as *SH D1* to *SH D3*). Degree of 3 gives the best performance as it has more capacity to model complicated light effects.

## 6 CONCLUSION

This paper introduces predictive 3D Gaussian splats, a lightweight representation that dramatically reduces storage for large-scale scenes compared to 3DGS, while maintaining high-fidelity rendering results. We propose an efficient structural representation, *i.e.*, *parent-children* structure to model the inherent spatial relationship among nearby splats. The *children* splats and most Gaussian attributes can be estimated during rendering by utilizing *parent*. Additionally, we leverage a hash grid and self-attention on aggregated features to enforce connectivity for *parent* and *children* nodes. We conduct extensive experiments on benchmark datasets to validate our design and demonstrate the our advantages of storage saving and high-quality novel view synthesis. For future work, since our representation is orthogonal to other compression techniques, we can combine it with methods like adaptive Gaussian pruning to achieve greater efficiency. This integration could enhance storage performance and reduce computational overhead, making our system more robust and scalable. Exploring these synergies will be a valuable direction for optimizing representation and improving overall efficiency in various applications.

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

# A APPENDIX

## A.1 VISUAL EXPLANATION OF ADAPTIVE TREE MANIPULATION

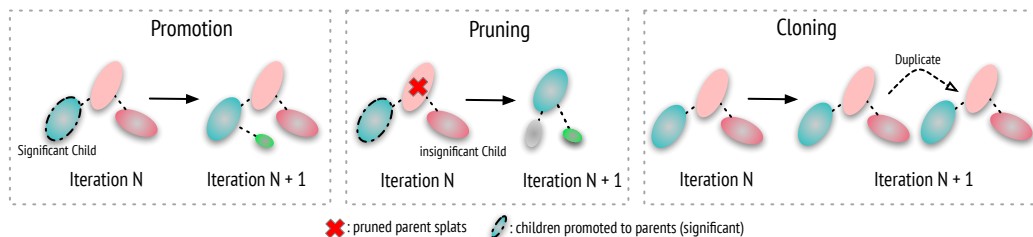

Figure 8: Demonstration of **Promotion**, **Pruning**, and **Cloning** operations in ATM.

As illustrated in Fig. 8, a *significant child* splat is promoted to the *parent* in the next iteration (*i.e.*, $N + 1$) if satisfying the criterion (*i.e.*, $\nabla_{pos}^c > \tau_{pos}^c$ ). In the iteration $N + 1$, the newly promoted *parent* has its own *children* splats (*e.g.*, the green node in the tree). When pruning the *parent* splats, *significant* splats (*e.g.*, the green node) and *insignificant* splats (*e.g.*, the red node) are handled differently, where the *insignificant* splats are pruned together with the *parent* whereas the *significant* splat turns to a new tree in iteration $N + 1$. Lastly, the cloning operates on the tree level where *children* splats are also cloned.

## A.2 IMPLEMENTATION DETAILS

In this section, we provide more details for our training. We first provide the hyper-parameters used during the training in A.2.1. We discuss the architecture and training details in A.2.2. Next we show the visual illustration and the implementation details of the contraction in Sec. A.2.3. Then, we analyze the effectiveness of the warm-up training strategy employed in our method in Sec. A.2.4 and the convergence in Sec. A.2.5. Lastly, we discuss the pre-filtering of *parent* points and its implementation in Sec. A.2.6.

### A.2.1 SETTINGS OF HYPER-PARAMETERS

We employ different learning schedules for different modules. For the hash grid, we start with a learning rate of $2e^{-3}$ and end with a rate of $2e^{-5}$. For opacity, we start with $1e^{-3}$ and end with $2e^{-5}$. The scale and rotation parameters utilize a constant learning rate of $1e^{-4}$. Additionally, we maintain a constant learning rate of $2e^{-4}$ for the attention module. We apply a standard exponential decay scheduling (Kerbl et al., 2023; Sara Fridovich-Keil and Alex Yu et al., 2022) to all modules.

### A.2.2 ARCHITECTURE AND TRAINING DETAILS

We use Instant-NGP (Müller et al., 2022b) as our hash-grid owing to its compact and efficient design and 2 layers MLP for all the MLPs. Following the practices in (Barron et al., 2022a; Müller et al., 2022b) we use scene contraction to map the position into $[0, 1]$ before feeding it to the Instant-NGP. This helps bring the splats that are occasionally outside the Axis-Aligned-Bounding-Box (AABB) due to the densification of splatsalong with the position updates. We estimate the AABB with the initial COLMAP (Schönberger & Frahm, 2016) point cloud. We set $\lambda = 0.5$ for all the experiments and train the model for $30K$ steps with $7.5K$ steps warm-up stage. In our experiments, we initially use lower resolution images to train the model $7.5K$ steps for warm-up, after which we transit to high-resolution images. More precisely, following the setting from 3DGS (Kerbl et al., 2023), we employ $32\times$ downsampling for the Mip-NeRF $360°$ dataset and $4\times$ downsampling for the Tank&Temples (Knapitsch et al., 2017) and Deep Blending (Hedman et al., 2018) datasets in the warm-up stage. The number of *children* splats ($K$) used in our experiment varies across scenes. We empirically found that 2 *children* are enough for most scenes and we believe this is because our sub-tree expansion allows the tree to grow deeper and compensate the breadth requirement. Please refer to A.4 material for details.

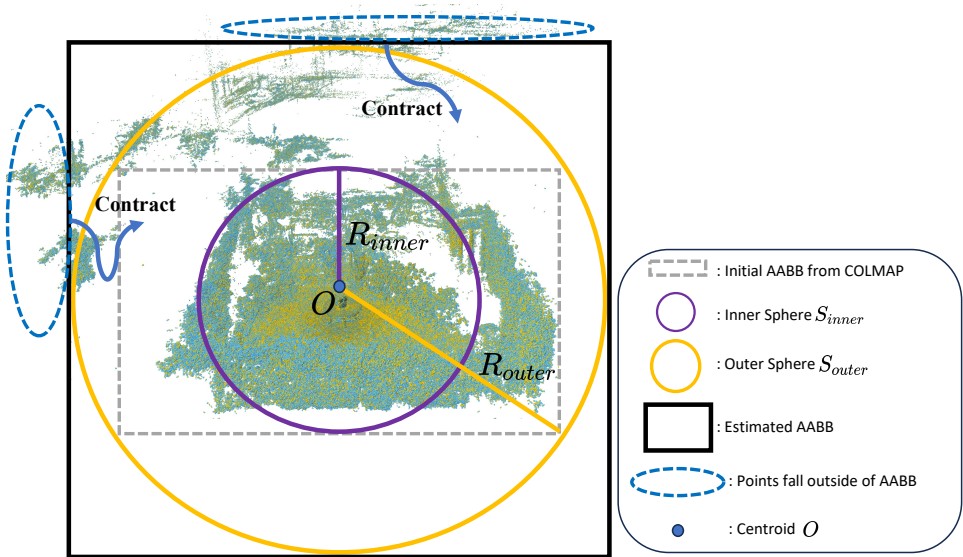

Figure 9: Illustration for our implemented contraction.

### A.2.3 DETAILS FOR CONTRACTION

We illustrate the process of contraction (described in Sec. 3.5 of the main paper) in Fig. 9 and Alg. 1. We calculate the inscribed and circumscribed spheres (*i.e.*, $S_{inner}$ and $S_{outer}$) with radius $R_{inner}$ and $R_{outer}$ of the initialized Axis-Aligned Bounding Box (AABB), which is estimated from the point cloud generated from COLMAP (Schönberger & Frahm, 2016). The estimated AABB is the circumscribed cube of the outer sphere $S_{outer}$. Points falling outside of the outer sphere are brought back to $S_{outer}$.

---

**Algorithm 1** AABB Estimation and Contraction

---

**Require:** Initialized AABB: $AABB_{init}$, point cloud: $PC$
    $S_{inner} \leftarrow$ Inscribed sphere of $AABB_{init}$                     ▷ centered at $O$
    $R_{inner} \leftarrow$ Radius of $S_{inner}$
    $S_{outer} \leftarrow$ Circumscribed sphere of $AABB_{init}$            ▷ centered at $O$
    $R_{outer} \leftarrow$ Radius of $S_{outer}$
    $AABB_{est} \leftarrow$ Circumscribed cube of $S_{outer}$

    **for** $p$ in $PC$ **do**                             ▷ contract the points
        **if** $\|p - O\| \leq R_{inner}$ **then**
            $p \leftarrow p$
        **else if** $\|p - O\| > R_{inner}$ **then**
            $p \leftarrow \left(R_{outer} - \frac{1}{\|p-O\|}\right)\left(\frac{p-O}{\|p-O\|}\right) + O$     ▷ infinity is on $S_{outer}$
        **end if**
    **end for**

---

### A.2.4 ANALYSIS OF WARM-UP

We run two experiments on the `Garden` (Barron et al., 2022a) scene in 10K steps to show the effectiveness of the warm-up in our method.

We have found that using warm-up in training with low resolution images at early stages helps the points populate the empty areas, especially when the COLMAP (Schönberger & Frahm, 2016) initialization is poor. Fig. 10 shows the point cloud and corresponding rendered images from different training approaches. As can be seen, the warm-up training (second row) has a better reconstruction and rendering quality for the background scene, which is poorly initialized from COLMAP (Schönberger & Frahm, 2016).

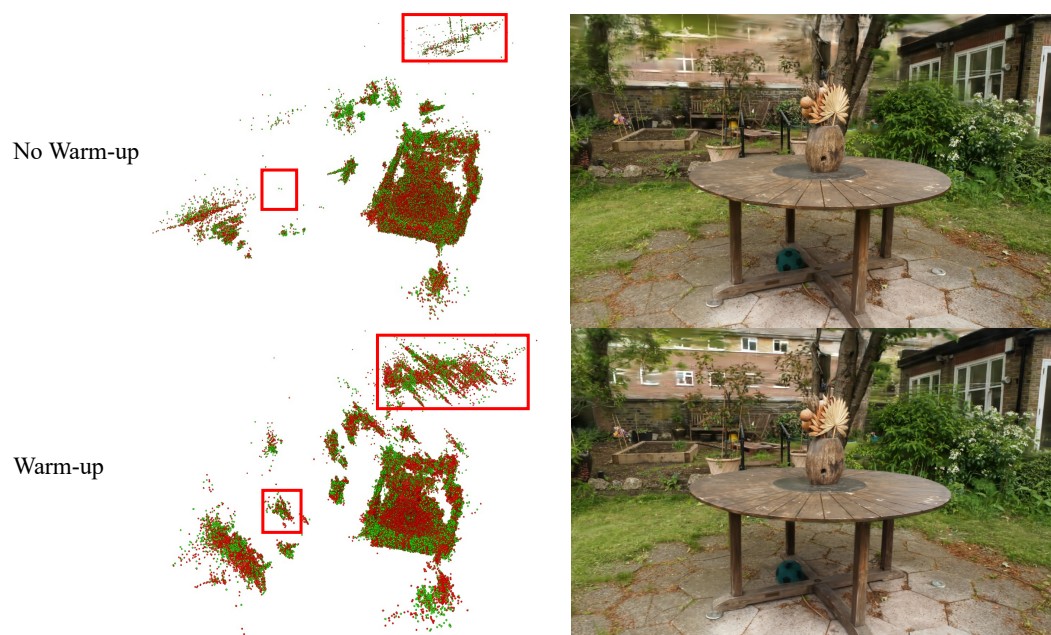

Figure 10: Analysis of warm-up. *First row*: training without warm-up. *Second row*: training with warm-up. *Left column*: points distribution. *Right column*: rendered images with the point cloud.

### A.2.5 ANALYSIS OF CONVERGENCE

Leveraging the hash-grid structure increases the per-step training time compared to 3DGS (Kerbl et al., 2023). For example, 3DGS requires approximately 23 minutes to reach a PSNR of 30.63 at 30K steps on scene room from MipNeRF-360 dataset (Barron et al., 2022a), our method takes about 27.6 minutes, and HAC (Chen et al., 2024) requires a similar training time of 27.1 minutes. Nonetheless, our approach strikes a good balance between size, speed, and quality. Furthermore, thanks to our inference phase optimization strategy in Sec. 5.1, the hash-grid structure has a minimal impact on inference time, allowing our approach to run in real-time on mobile devices.

### A.2.6 VIEW FRUSTUM CULLING

We apply pre-filtering on the *parent* points before querying features for attributes prediction by culling the view frustum with depth, leading to the computation reduction and the training speedup.

We empirically observe that 15% - 25% points are removed across scenes. The implementation is shown in Alg. 2.

---

**Algorithm 2** View Frustum Culling

---

**Require:** points $P : N \times 3$, view matrix $M : 4 \times 4$
    $P_{homo} \leftarrow Concat(P, ones)$                                    ▷ $P_{homo} : N \times 4$
    $P_{view} \leftarrow M * (P_{homo})^T$                           ▷ $*$ is matrix multiplication
    $mask \leftarrow P_{view}[2, :] > 0.201$                         ▷ depth $> 0.201$
    $P_{filtered} \leftarrow P[mask]$

---

### A.3 MORE ABLATION STUDY

**Scene Contraction.** We analyze the proposed contraction technique applied on the unbounded scene (Tab. 3 w/o Contract.). Compared with *Full*, we get inferior performance (0.8 PSNR drop), and tend to have training instability issues because the points occasionally move outside the Axis-Aligned Bounding box.

## A.4    Per-scene Quantitative Results

We provide the per-scene results on the benchmark datasets. Tab. 4 shows the results on the Mip-NeRF 360° dataset (Barron et al., 2022a). Tab. 5 demonstrates the results on the Tank&Temples dataset (Knapitsch et al., 2017) and the Deep Blending dataset (Hedman et al., 2018). We report the per-scene storage (in MB), the number of *parents* and *children*, and the metrics for image quality evaluation.

Table 4: Per-scene metrics for our approach on the Mip-NeRF 360° dataset Barron et al. (2022a).

|  | Metric | Garden | Bicycle | Stump | Room | Counter | Kitchen | Bonsai |
|---|---|---|---|---|---|---|---|---|
| Compact GS | PSNR | 26.81 | 24.77 | 26.46 | 30.88 | 28.71 | 30.480 | 32.08 |
|  | SSIM | 0.832 | 0.723 | 0.757 | 0.919 | 0.902 | 0.919 | 0.939 |
|  | LPIPS | 0.161 | 0.286 | 0.278 | 0.209 | 0.205 | 0.131 | 0.193 |
|  | Storage (MB) | 62.78 | 62.99 | 54.66 | 34.21 | 34.34 | 44.45 | 35.44 |
| Light GS | PSNR | 26.73 | 24.96 | 26.70 | 31.27 | 28.11 | 30.40 | 31.01 |
|  | SSIM | 0.836 | 0.738 | 0.768 | 0.926 | 0.893 | 0.914 | 0.944 |
|  | LPIPS | 0.155 | 0.265 | 0.261 | 0.220 | 0.218 | 0.147 | 0.204 |
|  | Storage (MB) | - | - | - | - | - | - | - |
| Scaffold GS | PSNR | 27.17 | 24.50 | 26.27 | 31.93 | 29.34 | 31.30 | 32.70 |
|  | SSIM | 0.842 | 0.705 | 0.784 | 0.925 | 0.914 | 0.928 | 0.946 |
|  | LPIPS | 0.146 | 0.306 | 0.284 | 0.202 | 0.191 | 0.126 | 0.185 |
|  | Storage (MB) | 271.00 | 248.00 | 493.00 | 133.00 | 194.00 | 173.00 | 258.00 |
| 3D GS | PSNR | 27.25 | 25.10 | 26.66 | 31.50 | 29.11 | 31.53 | 32.16 |
|  | SSIM | 0.856 | 0.747 | 0.756 | 0.925 | 0.914 | 0.932 | 0.946 |
|  | LPIPS | 0.122 | 0.244 | 0.243 | 0.198 | 0.184 | 0.117 | 0.181 |
|  | Storage (MB) | 1331.33 | 1350.78 | 1073.60 | 350.14 | 276.52 | 411.76 | 295.08 |
| Ours-C1 | PSNR | 27.17 | 24.32 | 25.75 | 31.62 | 28.54 | 30.47 | 31.32 |
|  | SSIM | 0.832 | 0.672 | 0.768 | 0.913 | 0.889 | 0.910 | 0.923 |
|  | LPIPS | 0.169 | 0.355 | 0.312 | 0.229 | 0.226 | 0.147 | 0.206 |
|  | Storage (MB) | 26.90 | 25.71 | 34.84 | 16.85 | 17.92 | 23.68 | 17.74 |
| Ours-C2 | PSNR | 27.38 | 24.78 | 26.41 | 31.82 | 28.75 | 30.71 | 32.14 |
|  | SSIM | 0.842 | 0.701 | 0.751 | 0.916 | 0.894 | 0.913 | 0.935 |
|  | LPIPS | 0.156 | 0.325 | 0.260 | 0.224 | 0.218 | 0.146 | 0.192 |
|  | Storage (MB) | 33.01 | 31.72 | 41.09 | 23.02 | 24.04 | 30.05 | 23.61 |
| Ours-C3 | PSNR | 27.63 | 24.90 | 26.43 | 31.84 | 29.10 | 31.27 | 32.67 |
|  | SSIM | 0.847 | 0.717 | 0.753 | 0.917 | 0.900 | 0.918 | 0.941 |
|  | LPIPS | 0.147 | 0.303 | 0.267 | 0.220 | 0.212 | 0.137 | 0.186 |
|  | Storage (MB) | 39.40 | 37.81 | 47.24 | 28.95 | 30.02 | 35.92 | 29.84 |
|  | # of Parents | 1.20M | 1.06M | 1.86M | 330K | 419K | 913K | 403K |
|  | # of Children ($k$) | 2 | 2 | 2 | 2 | 2 | 1 | 2 |

Table 5: Per-scene metrics for our approach on the Tank&Temples dataset Knapitsch et al. (2017) and the Deep Blending dataset Hedman et al. (2018).

|  | Metric | Tank&Temples | | Deep Blending | |
|---|---|---|---|---|---|
|  |  | Truck | Train | Drjohnson | Playroom |
| Compact GS | PSNR | 25.070 | 21.560 | 29.260 | 30.320 |
|  | SSIM | 0.871 | 0.792 | 0.9000 | 0.902 |
|  | LPIPS | 0.163 | 0.240 | 0.258 | 0.258 |
|  | Storage (MB) | 41.57 | 37.29 | 47.98 | 38.45 |
| Light GS | PSNR | 24.561 | 21.095 | - | - |
|  | SSIM | 0.855 | 0.760 | - | - |
|  | LPIPS | 0.188 | 0.296 | - | - |
|  | Storage (MB) | - | - | - | - |
| Scaffold GS | PSNR | 25.77 | 22.15 | 29.80 | 30.62 |
|  | SSIM | 0.883 | 0.822 | 0.907 | 0.904 |
|  | LPIPS | 0.147 | 0.206 | 0.250 | 0.258 |
|  | Storage (MB) | 107.00 | 66.00 | 69.00 | 63.00 |
| 3D GS | PSNR | 25.350 | 22.070 | 29.060 | 29.870 |
|  | SSIM | 0.878 | 0.812 | 0.899 | 0.901 |
|  | LPIPS | 0.148 | 0.208 | 0.247 | 0.247 |
|  | Storage (MB) | 608.70 | 255.82 | 773.61 | 553.03 |
| Ours-C1 | PSNR | 24.93 | 21.44 | 28.89 | 29.75 |
|  | SSIM | 0.856 | 0.763 | 0.894 | 0.895 |
|  | LPIPS | 0.196 | 0.283 | 0.280 | 0.284 |
|  | Storage (MB) | 23.11 | 20.90 | 23.59 | 22.21 |
| Ours-C2 | PSNR | 25.22 | 21.72 | 28.93 | 30.28 |
|  | SSIM | 0.862 | 0.777 | 0.902 | 0.902 |
|  | LPIPS | 0.184 | 0.272 | 0.287 | 0.268 |
|  | Storage (MB) | 30.73 | 27.36 | 29.84 | 28.46 |
| Ours-C3 | PSNR | 25.45 | 22.18 | 29.34 | 30.44 |
|  | SSIM | 0.866 | 0.792 | 0.898 | 0.905 |
|  | LPIPS | 0.182 | 0.240 | 0.270 | 0.265 |
|  | Storage (MB) | 36.01 | 34.63 | 35.80 | 35.00 |
|  | # of Parents | 1M | 900K | 900K | 834K |
|  | # of Children ($k$) | 1 | 1 | 2 | 2 |

