# OpenReview forum: "Lightweight Predictive 3D Gaussian Splats"
_ICLR.cc/2025/Conference — ICLR 2025 Poster_

### Official Review · Reviewer_9HaV · 2024-11-02

**Soundness:** 4
**Presentation:** 3
**Contribution:** 4
**Rating:** 8
**Confidence:** 3

**Summary:**

This paper proposes a storage-effective 3D gaussian splatting representation. It uses hierarchical tree structure use similarity between nearby 3D gaussians.  The tree structures are manipulated adaptively in the optimization processes. The method not only outperform SOTA methods, but it is also shown to run effectively in mobile devices.

**Strengths:**

Storage-effective representation for 3D gaussians.

The proposed method shown to outperform existing SOTA methods.

The proposed method shown to run on mobile devices.

Ablation study to show necessities of the proposed components.

**Weaknesses:**

There is only one figure (i.e., fig.2) for explaining the proposed method. Thus, it may not be easy to follow the process of the method.

**Questions:**

The format of the sub-section titles is not consistent. Some of them are ended with ":", some with "." and some with " ".

---

> ### Author Response · Authors · 2024-11-23
>
> We sincerely thank the reviewer for the positive feedback and suggestions. In the revised version, we have corrected the suggested typos and added an additional graph to explain our approach in Appendix A.1.

---

> ### Comment · Reviewer_9HaV · 2024-11-25
>
> The explanation in the appendix improves the cleanness of the tree manipulation. The feedbacks to my questions seem reasonable.
>
> I will keep my original rating.

---

> > ### Author Response · Authors · 2024-11-25
> >
> > Thank you for your quick response and positive rating! We're pleased to hear that your concerns have been addressed. We sincerely appreciate your time and valuable feedback.

---

### Official Review · Reviewer_zk78 · 2024-11-02

**Soundness:** 3
**Presentation:** 4
**Contribution:** 4
**Rating:** 8
**Confidence:** 4

**Summary:**

This paper presents a new representation for 3D Gaussian splats that drastically reduces disk space requirements while providing similar or improved quality compared to previous methods. The method detects inherent common features between splats in close proximity using an attention mechanism (ATM) and exploits these using a hierarchical tree structure in which only the parent splats need to be stored. As a very extensive evaluation on current benchmark data sets shows, the new representation achieves an average reduction in disk footprint of 20x compared to the original 3DGS, with improved PSNR and comparable SSIM and LPIPS, and a 2-5x reduction in storage compared to new works on more efficient representation. In addition, the method is presented to real rendering applications on mobile devices and AR glasses.
In contrast to the competing method CompactGS, it uses a combination of neural fields and self-attention layers to predict not only view-dependent colours but also geometric properties. Secondly, in contrast to competing approaches that store the position of each splat in the point cloud explicitly like ScaffoldGS, the method stores only a small subset of splats, referred to as parents, while the remaining points are predicted on-the-fly during rendering, significantly reducing memory requirements. In contrast to recently presented anchor-based methods, the hierarchical tree representation in combination with the proposed Adaptive Tree Manipulation takes into account the importance of both parent and child splats, which enables a subtree expansion strategy.

**Strengths:**

The idea of deriving the positions of child splats and associated attributes – position, color, scale, etc. – from the parent using a small neural network and only storing parent splats together with the weights of the neural network is new and should definitely be published.
The paper gives a very nice introduction to the problem and a very good overview of the existing approaches.
The method itself is explained in detail and is easy to follow. Comprehensive ablation studies show the influence of the different components of the algorithm on its performance. The extensive evaluation impressively demonstrates the strength of the method compared to the state of the art. It is interesting to note that, in contrast to the original 3DGS, the quality in some of the benchmark comparisons is better despite the considerable compression of the representation.

**Weaknesses:**

Unfortunately, there is no code to go with the paper, so that the procedure can be tested independently.

**Questions:**

In the  text to Figure 2 "patent" should be replaced by "parent".

Maybe you comment already in section 4.1. that in your code you have chosen $K$ to be 2?

---

> ### Author Response · Authors · 2024-11-23
>
> We sincerely thank the reviewer for the positive comments. We address the suggested typos in the revised version. We will release our code and pre-trained models upon the paper acceptance.
>
> Regarding the choice of *K*, we choose different K for different scenes. As our adaptive tree manipulation (ATM) can grow trees in both depth and breadth, we notice that setting *K* as 2 achieves the best results for most scenes. In Supplementary (Tab.4 and Tab.5), we show the value of *K* for different datasets.

---

### Official Review · Reviewer_ZQ2P · 2024-11-03

**Soundness:** 3
**Presentation:** 3
**Contribution:** 2
**Rating:** 5
**Confidence:** 5

**Summary:**

This paper proposes a lightweight 3D Gaussian framework that models inherent spatial relationships within a hierarchical tree structure. To optimize the tree structure, the authors suggest adaptive tree manipulation (ATM), an adaptive growth and pruning strategy for parent and children nodes. Consequently, it demonstrates efficient storage size and superior rendering quality compared to compact 3D Gaussian representations, e.g., LightGS, CompactGS, and EAGLES.

**Strengths:**

- The proposed tree structure enables parent nodes to represent children nodes using an on-the-fly decoding pipeline. This results in the small disk usage of this representation with high rendering quality.

- The optimization schemes, ATM for growth and warm-up for initial training, lead to achieving stable and effective optimization of the proposed representation.

**Weaknesses:**

- The main weakness of this paper is the limited technical novelty. The proposed tree structure mainly comes from the anchor-based representation, Scaffold-GS [1]. Also, the adaptive manipulation of children nodes is proposed in HAC [2], with more efficient learnable masks. Moreover, the usage of a hash grid for efficient 3DGS representation is also proposed in HAC and CompactGS [3].

- Also, it demonstrates a slower rendering speed compared to 3DGS, as we can see in L458. It indicates that this approach requires more 3D Gaussians splats (both parent splats and children splats) for the rendering phase and only consumes less disk usage in the storage. Therefore, the claim for **"lightweight"** representation seems overclaimed.

- Furthermore, there is limited comparison to existing compact representations. Recently, many papers have been addressing the inefficiency issue of 3DGS. However, there are too less methods in the comparison. Please refer to the question section for the related approaches.

---
**Reference**
1. Lu et al., Scaffold-GS: Structured 3D Gaussians for View-Adaptive Rendering, CVPR 2024
2. Chen et al., HAC: Hash-grid Assisted Context for 3D Gaussian Splatting Compression, ECCV 2024
3. Lee et al., Compact 3D Gaussian Representation for Radiance Field, CVPR 2024

**Questions:**

- The results of CompactGS in Table 1 seem to be not applying post-processing, such as quantization of the hash grid. As this paper reports the storage size for the quantized hash grid as described in L443, it is fair to report the post-processed results of CompactGS.

- Moreover, it seems that the tree structure and hash grid design require more training time compared to 3DGS. Please provide the training time to optimize this representation.

- Also, the detailed number of 3D Gaussians for 3DGS should be added to prove that this method requires less number of Gaussians as described in the quantitative results section.

- Furthermore, it is recommended to add more comparisons for the compact 3DGS representations as below. According to the reviewer guideline, the authors do not have to compare papers which are recently published in ECCV'24. But, as the idea this paper is significantly related to HAC [1], it could be helpful for comparing with HAC in terms of both performance and methodology.

  - Compressed3D [2] (CVPR 2024)
  - Reduced3DGS [3] (I3D 2024)
  - HAC [1] (ECCV 2024)
  - SOG [4] (ECCV 2024)
  - Compact3D [5] (ECCV 2024)

---
**Reference**
1. Chen et al., HAC: Hash-grid Assisted Context for 3D Gaussian Splatting Compression, ECCV 2024
2. Niedermayr et al., Compressed 3D Gaussian Splatting for Accelerated Novel View Synthesis, CVPR 2024
3. Papantonakis et al., Reducing the Memory Footprint of 3D Gaussian Splatting, I3D 2024
4. Morgenstern et al., Compact 3D Scene Representation via Self-Organizing Gaussian Grids, ECCV 2024
5. Navaneet et al., Compact3D: Smaller and Faster Gaussian Splatting with Vector Quantization, ECCV 2024

---

> ### Author Response · Authors · 2024-11-23
>
> **Q1. Clarification on Novelty.**
>
> We would like to thank the reviewer for suggesting the concurrent work (*e.g.*,  HAC [1], Compressed3D [2], Reduced3D [3], SOG [4], and Compact3D [5]). They are indeed relevant and we will discuss them in the revised paper. In the following, we would like to kindly emphasize the major differences between our method and the concurrent and suggested works.
>
> 1. *Ours v.s. Anchor-based Methods*. As discussed in the main paper (Line 142-145 and Line 150-154), there are two major differences between anchor-based approaches (*e.g.*, ScaffoldGS [10]) and our approach. First, the anchor structure in ScaffoldGS [10] is essentially a reparameterization of the 3DGS [9] representation, where splat positions are parameterized relative to anchors. Consequently, ScaffoldGS [10] must store both the anchors and the relative positions during inference, which results in a lower compression rate, as shown in Table 1 of its paper. In contrast, our hierarchical tree structure is predictive, leveraging spatial relationships via the tree combined with an attention mechanism to enforce similarity. This allows us to store only the parent splats, achieving significantly higher compression rates (as mentioned by Reviewers zk78, 9HaV). Second, the proposed hierarchical tree structure fundamentally differs from the anchor-based structure. Unlike anchors, which only capture a single level of connectivity between splats, our tree representation combined with the proposed ATM  takes into account the significance of both parent and children splats, allowing for a strategy of sub-tree expansion.
>
>     HAC [1] closely follows the anchor-based method ScaffoldGS [10]. The underlying representation (*i.e.*, anchor) is essentially the same as ScaffoldGS [10] (the discussion above between anchors and hierarchical tree representation still apply). Moreover, our ATM is different from the learnable masks in HAC [1] in many aspects.  ATM contains three operations: Promotion, Pruning and Cloning (Line 283-287), introducing an effective way to expand and delete the trees. The learnable mask in HAC [1] is similar to the Gaussian volume mask in CompactGS [8], which prunes the unimportant individual splats with binary masks without considering the anchors as a whole. Our ATM is the key to enabling the hierarchical tree in our framework.
> 2. *Ours v.s. Hash-grid based Methods*. Our method uses hash-grids (displacement and attributes) to query features for both positions and geometric attributes (Line 147-148, Figure 2) and predict the children's positions and attributes on-the-fly, whereas the CompactGS [8] only predicts the positions. To facilitate such workflow, we adopt a specific design of hash-grid. We assume the features of positions and attributes of splats should not be disentangled in the spatial grid. Thus, we propose a strategy to first query the positions and re-query the hash-grid with the predicted positions for their geometric attributes. Our use of a hash-grid combined with the proposed attention mechanism is to ensure the inherent feature sharings between the splats in their vicinity. We kindly ask the reviewer to check Section 4.1 - Predicting Position for more detailed explanation.
>
> Thanks again for the suggestions. We add an explanation to the revised version in Appendix A.2.
>
> ---
>
> **Q2. About Post-processed Metrics.**
>
> We thank the reviewer for suggesting on reporting the post-processed metrics of CompactGS [8]. We include the table for the MipNeRF-360 [7] dataset below. To have the same setting between our work and CompactGS [8], we remove the Huffman encoding and pruning accordingly from the CompactGS [8]. As can be seen, our model achieves smaller storage and better reconstruction than CompactGS [8].
>
> >| Scene        | PSNR   | SSIM   | LPIPS  | Storage (MB) |       | PSNR   | SSIM   | LPIPS  | Storage (MB) |
> >|--------------|--------|--------|--------|--------------|-------|--------|--------|--------|--------------|
> >| **CompactGS**|        |        |        |              | **Ours-C2**|        |        |        |              |
> >| bicycle      | 24.75  | 0.723  | 0.286  | 47.62           | | 24.78  | 0.701  | 0.325  | 31.72        |
> >| garden       | 26.77  | 0.831  | 0.161  | 48.46        || 27.38  | 0.842  | 0.156  | 33.01     |
> >| stump        | 26.40  | 0.755  | 0.280  | 39.03       ||  26.41  | 0.751  | 0.260  | 41.09          |
> >| room         | 30.88  | 0.918  | 0.209  | 20.21            || 31.82  | 0.916  | 0.224  | 23.02        |
> >| counter      | 28.68  | 0.901  | 0.205  | 20.43           | |28.75  | 0.894  | 0.218  | 24.04        |
> >| kitchen      | 30.48  | 0.919  | 0.131  | 29.59           | |30.71  | 0.913  | 0.146  | 30.05        |
> >| bonsai       | 32.03  | 0.938  | 0.193  | 21.60           | |  32.14  | 0.935  | 0.192  | 23.61           |
> >| **Average**  |   28.57     | 0.855       |  0.210  | 32.42            | |    28.86 | 0.851 | 0.217| 29.50|

---

> ### Author Response · Authors · 2024-11-23
>
> **Q3. About Training Time.**
>
> We thank the reviewer for the suggestion and will incorporate a discussion on convergence speed in our revised paper. Leveraging the hash-grid structure increases the per-step training time compared to 3DGS [9]. For example, 3DGS [9] requires approximately 23 minutes to reach a PSNR of 30.63 at 30K steps on scene room from MipNeRF-360 [7] dataset, our method takes about 27.6 minutes, and HAC [1] requires a similar training time of 27.1 minutes. Nonetheless, our approach strikes a good balance between size, speed, and quality. Furthermore, thanks to our inference phase optimization strategy (Line 452 of main paper), the hash-grid structure has a minimal impact on inference time, allowing our approach to run in real-time on mobile devices.
>
> Thanks again for the suggestions. In the revised version, we explain the training convergence in Appendix A.3.5.
>
> ---
>
> **Q4. More Explanation for Lightweight.**
>
> We thank the reviewer for the feedback. We have reported the number of parent and child splats in our Supplementary. For instance, our approach requires an average of 2.52M total splats on the MipNeRF-360 [7] dataset, compared to 3.07M splats required by 3DGS[9]. The inference overhead primarily arises from the first-frame feature queries and the MLP used for color computation during per-frame rendering (Lines 452–457). We appreciate the reviewer’s suggestion and will clarify the term "lightweight" more explicitly in our revised paper.
>
> ---
>
> **Q5. Comparison to Concurrent Works.**
>
> We sincerely thank the reviewer for the suggestion. While it is not necessary to compare our work with ECCV 2024 papers, we aim to address the reviewer’s concerns comprehensively. To this end, we provide quantitative comparisons between our approach and concurrent works on two unbounded scenes from the MipNeRF-360 [7] dataset. Please kindly note that the results for Compact3D [5] have already been discussed and are presented in Table 1 of our paper (Line 390), so they are not included in the comparisons below.
>
>
> >| Metric | Compressed3D |HAC|Reduced3DGS|Ours-C3 |
> >|----------------|----------|-------|-----|--------------|
> >| PSNR |25.86 | 26.27 | 26.04 |26.27 |
> >| SSIM | 0.797 | 0.796 |0.807 | 0.782 |
> >| LPIPS | 0.192 | 0.199| 0.170  |0.225  |
> >|Storage (MB) |46.85 |  40.14 | 47.50 |38.60 |
>
> Our approach achieves the smallest storage requirements while maintaining comparable or superior PSNR compared to various concurrent works. Notably, we require less storage than HAC [1], even though HAC [1] leverages BiRF [6] to achieve 1-bit quantization of the grid features, whereas our method uses 8-bit. These results demonstrate the efficacy of our tree representation compared to anchor-based methods. Since quantization techniques are orthogonal to our representation, we anticipate achieving an even higher compression ratio by adopting the same settings as HAC [1].
>
> ---
>
> **References**
> 1. Chen et al., HAC: Hash-grid Assisted Context for 3D Gaussian Splatting Compression, ECCV 2024
> 2. Niedermayr et al., Compressed 3D Gaussian Splatting for Accelerated Novel View Synthesis, CVPR 2024
> 3. Papantonakis et al., Reducing the Memory Footprint of 3D Gaussian Splatting, I3D 2024
> 4. Morgenstern et al., Compact 3D Scene Representation via Self-Organizing Gaussian Grids, ECCV 2024
> 5. Navaneet et al., Compact3D: Smaller and Faster Gaussian Splatting with Vector Quantization, ECCV 2024
> 6. Shin et al., Binary Radiance Field, NeurIPS 2024
> 7. Barron et al., Mip-NeRF 360: Unbounded Anti-Aliased Neural Radiance Fields, CVPR 2022
> 8. Lee et al., Compact 3D Gaussian Representation for Radiance Field, CVPR 2024
> 9. Kerbl et al., 3D Gaussian Splattingfor Real-Time Radiance Field Rendering, SIGGRAPH 2023
> 10. Lu et al., Scaffold-GS: Structured 3D Gaussians for View-Adaptive Rendering, CVPR 2024

---

> > ### Author Response · Authors · 2024-11-27
> >
> > Dear Reviewer ZQ2P,
> >
> > Thank you for your valuable feedback on our submission.
> >
> > We have provided additional explanations to address the key points you raised, including the fundamental differences in the methodology from the concurrent works, post-processing metrics, analysis of convergence, clarification of the term *lightweight* and additional discussion and comparison with ECCV 2024 papers. We would like to kindly ask if our responses sufficiently clarify your concerns or if there are any remaining issues you would like us to address. We appreciate your time and consideration.
> >
> > Best,
> >
> > Authors

---

> > ### Comment · Reviewer_ZQ2P · 2024-11-27
> >
> > I appreciate the authors’ responses to address my concerns in the discussion period. Despite the efforts, I still maintain my rating due to the limited rendering performance. I also agree that the technical novelty of this work is feasible, but the limited performance of this method is the main reason for my rating.
> >
> > **[Q1, Q3, Q4]**
> >
> > Thank you for the detailed explanations.
> >
> > **[Q2]**
> >
> > First of all, I think that it is fair to compare CompactGS with post-processed results since CompactGS exploits a hash grid only for color representation, while this paper also represents other attributes and children splats using a hash grid. Thus, CompactGS would show more robust performance according to the post-processing for the hash grid.
> > In the table, CompactGS shows comparable storage size with high rendering quality in SSIM and LPIPS. Therefore, this method does not achieve better reconstruction than CompactGS. Which components result in lower rendering quality in terms of SSIM and LPIPS?
> >
> > **[Q5]**
> >
> > Compared to existing works, it might seem that the rendering quality of this paper is better due to the high PSNR score. However, when we focus on the other metrics, SSIM and LPIPS, this paper shows significantly less performance than all of the suggested methods, including Compressed3D and Reduced3DGS. Note that Compressed3D (CVPR 2024) and Reduced3DGS (I3D 2024) can be requested for comparison, not concurrent works. Also, I wonder which scenes were used in the experiments. What are the results for Compressed3D and Reduced3DGS on other scenes?
> >
> >
> > If the authors resolve my additional questions regarding the rendering quality, I will consider raising my rating.

---

> > > ### Author Response · Authors · 2024-11-28
> > >
> > > We would like to thank the reviewer for recognizing and acknowledging the technical **novelty** of this work. In the following, we provide more explanation to help alleviate the remaining concern from the reviewer.
> > >
> > > **Q2. Clarification of Render Quality.**
> > >
> > > To address the question on the rendering quality, we also include the rendering metrics (*i.e.*, PSNR, SSIM, LPIPS)  on MipNeRF-360 dataset of  **Ours-C3** in the following table.
> > >
> > >
> > > | Methods / Avg. Metrics       | PSNR   | SSIM   | LPIPS |
> > > |--------------|--------|-----------------|-----------------------------------|
> > > | **CompactGS**|28.57     | 0.855 |  0.210  |
> > > | **Ours-C3**      | 29.11    | 0.857 |  0.210  |
> > >
> > > As demonstrated from the table, our method achieves more *robust* rendering quality compared to CompactGS in terms of *PSNR* (*i.e.*, 29.11 v.. 28.57) and *SSIM* (*i.e.*, 0.857 v.s 0.855) , while maintaining on-par *LPIPS* performance (*i.e.*, 0.210 for both).  These results quantitatively show the superior rendering capabilities of our method.
> > >
> > > Most importantly, we have included the visual comparison between our work and CompactGS in the Fig.4 of the main paper and our [demo page](https://anonymous0submissions.github.io/LPGS/). It is evident that our method achieves better visual quality across all scenes. For instance, in the garden scene, CompactGS struggles to render background regions clearly, particularly those far from the camera (*e.g.*, walls and windows). A similar issue is observed in the room scene, specifically with the curtain details. We suspect these limitations arise from the reliance of the hash-grid without a mechanism to explicitly exploit the structures between splats. Therefore, CompactGS may fail to recover the details of complex scenes, or to model the regions that are far from the foreground (possible due to how the covariance matrix of Gaussian is initialized for SfM in 3DGS, *i.e.*, distance between nearby points). Our method directly addresses these issues through its hierarchical tree representation and distance-aware modeling of Gaussians (Line 257). We kindly ask the reviewer to refer to the side-by-side visual comparisons for further clarity. Lastly, we emphasize the flexibility offered by our work, which includes multiple configurations (Tab. 1) to suit different performance and storage requirements. Our approach strikes a balanced compromise between rendering quality and storage efficiency.
> > >
> > > We hope the results presented in the table above, along with the visual comparisons, effectively address the reviewer’s concerns regarding rendering quality.

---

> ### Author Response · Authors · 2024-11-28
> **Q5 - Part 1**
>
> **Q5**. We thank the reviewer’s kindness and willingness to raise the score. In addition to the metrics, we provide a concise discussion comparing the methodologies of existing works (*e.g.*, Compressed3D and Reduced3DGS) and our approach.
>
> 1. Compressed3D employs a fundamentally different approach compared to existing methods in the field. Instead of integrating compression techniques during the training or reconstruction phase, it operates as a post-processing framework. This requires an already trained or reconstructed scene to begin with. Subsequently, a series of techniques, such as vector quantization (VQ) codebooks, fine-tuning, and dictionary-based compression like LZ77, are applied to achieve compression.  The reliance on a fully reconstructed scene as input distinguishes it from methods that incorporate compression techniques directly into the scene's modeling process. Techniques like LZ77 are orthogonal and applicable to many representation-based approaches.
> 2. Reduced3DGS closely follows the methodology of 3DGS but introduces additional techniques to enhance efficiency and storage. Specifically, it leverages pruning strategies to remove unnecessary splats, thereby reducing redundancy in the representation. During optimization, it adaptively selects the appropriate bands of spherical harmonics to balance detail retention and computational cost. Following this optimization, codebook-based quantization is applied to further compress the scene representation
>
> Our method is a representation-based approach that leverages a hierarchical representation of spats to model the spatial relationships which disjoint with post-processing based methods. More importantly, we demonstrate the real-world rendering applications on mobile devices while other methods are unclear (as not illustrated).

---

> > ### Author Response · Authors · 2024-11-28
> > **Q5 - Part 2**
> >
> > To address the concerns on the rendering quality, we include the rendering quality metrics in the following table on the rest of MipNeRF-360 dataset besides the garden and bicycle scenes reported previously.
> >
> >
> >
> > | Methods / Avg. Metrics   |PSNR | SSIM   | LPIPS |
> > |--------------|--------|-----------------|-------------------------|
> > | **Compressed3D**|29.54 | 0.882 |  0.211  |
> > | **Reduced3DGS**   | 29.59  | 0.887 |  0.202  |
> > | **Ours-C3**   |   30.26 | 0.887 |  0.204  |
> >
> > Our approach demonstrates better performance in PSNR, achieving the highest value among the compared methods. It also delivers an on-par SSIM with Reduced3DGS and surpasses Compressed3D in this metric. In terms of LPIPS, our method attains a comparable score with Reduced3DGS (*i.e.*, 0.204 v.s 0.202), while still outperforming Compressed3D (*i.e.*, 0.204 v.s 0.211). To better understand what these numbers signify, it's essential to discuss the focus of each metric. PSNR, SSIM, and LPIPS are widely used in reconstruction tasks, yet they emphasize different aspects of image quality.
> > - PSNR measures pixel-level fidelity, providing a direct and reliable indicator of how accurately the rendered image matches the ground truth. This metric is particularly sensitive to noise and compression artifacts that affect fine details.
> > - SSIM captures perceptual quality by evaluating luminance, structure, and contrast. While it provides a nuanced view of image quality, it can be overly sensitive to minor variations in contrast and brightness that may not significantly impact the perceived quality of the rendering.
> > - LPIPS focuses on perceptual similarity, emphasizing plausibility over pixel-perfect fidelity. While this makes it useful for assessing overall coherence, it can sometimes classify images as perceptually similar even when they deviate numerically from the ground truth.
> >
> > Methods leveraging compression techniques such as codebook-based (*e.g.*, Compressed3D, Reduced3DGS), grid-based (*e.g.*, our method, CompactGS), or spherical harmonics compression (*e.g.*, ScaffoldGS) might slightly degrade SSIM and LPIPS scores. For **grid-based** methods, this degradation can often be attributed to the interpolation of grid features. Such interpolation may result in softer, less defined boundaries in the reconstructed output, which might not align well with the sensitivity of SSIM and LPIPS to sharp structural details and perceptual plausibility. However, this interpolation simultaneously fosters feature sharing between splats, which plays a crucial role in accurately modeling local regions. This trade-off highlights the strength of grid-based approaches in capturing localized coherence, even if it slightly impacts these perceptual metrics (*i.e.*, SSIM and LPIPS).
> >
> > Moreover, the degradation of perceptual metrics does not necessarily equate to lower rendering quality. High-quality rendering often hinges on the ability to capture fine details and textures, which PSNR effectively quantifies.
> >
> > We hope the table above, in addition to our explanation of the metrics, addresses the reviewer’s questions on the performance of our method. We sincerely appreciate the reviewer’s time and effort in this reviewer-author discussion.

---

> > > ### Author Response · Authors · 2024-12-02
> > >
> > > Dear Reviewer ZQ2P,
> > >
> > > We would like to thank you again for your valuable feedback on our paper. We have included additional experiments and clarifications to address your concerns.
> > >
> > > As the period for the reviewer-author discussion is closing very soon, we would like to use this opportunity to kindly ask if our responses sufficiently clarify your concerns. We sincerely appreciate your time and consideration.
> > >
> > > Best Regards,
> > > Authors

---

### Author Response · Authors · 2024-11-23

We sincerely appreciate your time and effort in reviewing our submission and providing valuable and positive feedback to enhance our work. We are delighted that the reviewers have acknowledged the strength of our paper, including:

**novel hierarchical tree structure and representation** leading to small-disk usage with high rendering quality (Reviewer ZQ2P), storage-effective for 3D gaussians (Reviewer 9HaV), and is new and should definitely be published (Reviewer zk78);

**extensive experiments** outperforming existing SOTA methods (Reviewer zk78, ZQ2P) with efficient storage size and superior rendering quality (Reviewer 9HaV), and better than 3DGS on some benchmarks despite the considerable compression of the representation (Reviewer zk78);

**comprehensive ablation** showing the influence and necessities of different components from the proposed algorithm (Reviewer zk78,  9HaV);

**deployment of real rendering** demonstrating our model can run effectively on mobile devices and AR glasses (Reviewer zk78,  9HaV);

**nice introduction, good overview, and detailed explained method** of the paper (Reviewer zk78).

We also **revise the manuscript** with the following updates suggested from reviewers, including:
1. adding a visual explanation to illustrate our method in *Appendix A.1*;
2. adding a discussion of related work (*e.g.*, HAC [1]) in  *Appendix A.2*;
3. adding the analysis of training convergence in *Appendix A.3.5*;
4. fixing the typos in caption of Fig. 2;
5. fixing the typos in subsection titles.

The added content is highlighted in light blue color.

In the following, we provide detailed responses to each reviewer.

---

**Reference**
1. Chen et al., HAC: Hash-grid Assisted Context for 3D Gaussian Splatting Compression, ECCV 2024

---

### Meta-Review · Area_Chair_XFQa · 2024-12-18

**Metareview:**

The paper introduces a novel hierarchical 3DGS representation that effectively reduces storage requirements while maintaining similar or improved rendering quality compared to the standard 3DGS. By leveraging on inherent feature sharing among proximate splats, the method represents nearby splats using a reduced number of parent splats, leading to a more efficient and compact representation. Experimental results demonstrate that the proposed approach significantly decreases the storage footprint relative to existing or baseline 3DGS methods, without compromising rendering quality. Therefore, I recommend its acceptance.

**Additional Comments On Reviewer Discussion:**

Initially, the reviewers expressed concerns about the technical novelty and the rendering quality of the paper. During the rebuttal phase, the authors provided comprehensive explanations and additional experiments that addressed these issues. Reviewer ZQ2P had initially assigned a negative score to the paper. After several rounds of discussion, some of their concerns were alleviated, but ultimately the reviewer did not adjust their score or respond further to the authors. While the concern regarding the paper's contribution seems to have been resolved, Reviewer ZQ2P did not provide an updated opinion on the rendering quality. Given that both the qualitative and quantitative results presented by the authors are positive and the final review scores, I recommend accepting this paper as an spotlight paper.

---

### Decision · Program_Chairs · 2025-01-22

Accept (Poster)